# Latitudinal trends in stable isotope signatures and carbon concentrating mechanisms of northeast Atlantic rhodoliths

Laurie C. Hofmann[1], Svenja Heesch[2]

[1]Microsensor Group, Max Planck Institute for Marine Microbiology, Bremen, 28359, Germany
[2]Centre National de Recherche Scientifique, UMR 8227, Station Biologique de Roscoff, Roscoff, 29680, France

*Correspondence to*: Laurie C. Hofmann (lhofmann@mpi-bremen.de)

**Abstract.** Rhodoliths are free-living calcifying red algae that form extensive beds in shallow marine benthic environments (< 250 m), which provide important habitats and nurseries for marine organisms and contribute to carbonate sediment accumulation. There is growing concern that these organisms are sensitive to global climate change, yet little is known about their physiology. Considering their broad distribution along most continental coastlines, their potential sensitivity to global change could have important consequences for the productivity and diversity of benthic coastal environments. The goal of this study was to determine the plasticity of carbon concentrating mechanisms (CCMs) of rhodoliths along a latitudinal gradient in the Northeast (NE) Atlantic using natural stable isotope signatures. The $\delta^{13}C$ signature of macroalgae can be used to provide an indication of the preferred inorganic carbon source ($CO_2$ vs. $HCO_3^-$). Here we present the total ($\delta^{13}C_T$) and organic ($\delta^{13}C_{org}$) $\delta^{13}C$ signatures of NE Atlantic rhodoliths with respect to changing environmental conditions along a latitudinal gradient from the Canary Islands to Spitsbergen. The $\delta^{13}C_T$ signatures (-11.9 to -0.89) of rhodoliths analysed in this study were generally higher than the $\delta^{13}C_{org}$ signatures, which ranged from -25.7 to -2.8. We observed a decreasing trend in $\delta^{13}C_T$ signatures with increasing latitude and temperature, while $\delta^{13}C_{org}$ signatures were only significantly correlated to DIC. These data suggest that high latitude rhodoliths rely more on $CO_2$ as an inorganic carbon source, while low latitudes rhodoliths likely take up $HCO_3^-$ directly, but none of our specimens had $\partial^{13}C_{org}$ signatures less than -30, suggesting that none of them relied solely on diffusive $CO_2$ uptake. However, depth also has a significant effect on both skeletal and organic $\delta^{13}C$ signatures, suggesting that both local and latitudinal trends influence the plasticity of rhodolith inorganic carbon acquisition and assimilation. Our results show that many species, particularly those at lower latitudes, have carbon concentrating mechanisms that facilitate $HCO_3^-$ use for photosynthesis. This is an important adaptation for marine macroalgae, because $HCO_3^-$ is available at higher concentrations than $CO_2$ in seawater, and this becomes even more extreme with increasing temperature. The flexibility of CCMs in northeast Atlantic rhodoliths observed in our study may provide a key physiological mechanism for potential adaptation of rhodoliths to future global climate change.

## 1 Introduction

Rhodoliths are free-living calcifying red algae that form extensive beds in shallow marine benthic environments (< 250 m) which provide important habitats and nurseries for marine organisms and contribute to carbonate sediment accumulation. There is growing concern that these organisms are sensitive to global climate change (Hofmann and Bischof, 2014; McCoy and Kamenos, 2015), particularly ocean acidification. Ocean acidification may be detrimental to rhodoliths by reducing calcification rates or increasing dissolution rates (Hofmann and Bischof, 2014), which would have severe impacts on the communities supported by rhodolith beds and coastal carbonate accumulation. Considering their global distribution, it is important to understand how rhodoliths may be affected by changing environmental conditions, and if they have physiological mechanisms that will allow them to adapt. The response of marine macroalgae to ocean acidification may be closely linked to their inorganic carbon uptake mechanisms (Cornwall et al., 2017; Hepburn et al., 2011). Marine

macrophytes have diverse physiological mechanisms for concentrating $CO_2$ (carbon concentrating mechanisms: CCMs) that allow them to overcome the low concentration of $CO_2$ in seawater relative to $HCO_3^-$ by direct uptake of $HCO_3^-$ or enzymatic conversion of $CO_2$ to $HCO_3^-$ via carbonic anhydrase (Giordano et al., 2005). The species of inorganic carbon ($CO_2$ or $HCO_3^-$) taken up by marine macroalgae influences the stable carbon isotope signature of the tissue (Maberly et al., 1992; Raven et al., 2002). Therefore, the ratio of stable carbon isotopes ($\partial^{13}C$) in macroalgal tissue can be used as an indicator of whether or not $HCO_3^-$ is being used (Raven et al., 2002) using the formula

$$\partial^{13}C = \frac{^{13}C/^{12}C_{sample}}{^{13}C/^{12}C_{PDB}},$$ (1)

where $^{13}C/^{12}C$ is the ratio of the natural abundance of the carbon stable isotopes in the macroalgal tissue sample and carbonate from the Cretaceous Pee-Dee formation (PDB). Values greater than -10 ‰ indicate that a CCM is present and the macroalga is able to take up $HCO_3^-$, while values less than -30 ‰ indicate there is no CCM present and the macroalga relies solely on diffusive $CO_2$ uptake. Values in between -30 and -10 ‰ indicate uptake of both $CO_2$ and $HCO_3^-$.

Many studies have investigated the presence/absence of CCMs using $\partial^{13}C$ signatures across taxonomic groups and environmental gradients such as depth/light, $CO_2$, and latitude (Hepburn et al., 2011; Moulin et al., 2011; Raven et al., 2011; Stepien, 2015; Stepien et al., 2016), but so far few studies have investigated the flexibility of CCMs in a single species or group of marine macroalgae (Cornelisen et al., 2007; Cornwall et al., 2017; Mackey et al., 2015). Therefore, we investigated the plasticity of CCMs in *Lithothamnion* spp. and other rhodolith species across a latitudinal gradient from the Canary Islands to Spitsbergen using natural stable isotope signatures.

## 2 Materials and Methods

### 2.1 Sample Collection and Treatment

Rhodolith samples were collected at each site from 3-10 m depth via snorkelling or SCUBA-diving, with the exception of the samples from Mosselbukta, which were collected during the MSM55 cruise with the manned submersible JAGO between at 11, 25 and 40 m depth. Samples were air-dried or dried at 60°C for 48 hours. After drying, samples were ground to a powder in stainless steel shaking flasks with two chromium steel grinding balls in a micro-dismembrator ball mill for 30 s at 2000 rpm (B. Braun Biotech International, Melsungen, Germany). For $\delta^{13}C_T$ analysis, approximately 10 mg of powder was packed in a 5 x 9 mm tin capsule (HEKAtech GmbH, Wegberg, Germany) for each sample and sent to the UC Davis stable isotope facility. For $\delta^{13}C_{org}$ analysis, the rhodolith powder was treated with 300 µL 1 M HCl in 10.5 x 9 mm silver capsules (HEKAtech GmbH, Wegberg, Germany) to dissolve all inorganic carbon. The samples were left for several days in a fume hood until the HCl evaporated and all inorganic carbon was dissolved. The organic fraction was then washed with distilled water and re-dried before the capsules were packed and sent to the UC Davis stable isotope facility, where the samples were analyzed for $^{13}C$ and $^{15}N$ isotopes using a PDZ Europa ANCA-GSL elemental analyzer interfaced to a PDZ Europa 20-20 isotope ratio mass spectrometer (Sercon Ltd., Cheshire, UK). Samples were combusted at 1000°C in a reactor packed with chromium oxide and silvered copper oxide. Following combustion, oxides were removed in a reduction reactor (reduced copper at 650°C). $N_2$ and $CO_2$ were separated on a Carbosieve GC column (65°C, 65 mL/min) before entering the IRMS. During analysis, samples were interspersed with several replicates of different laboratory standards (glutamic acid and peach leaves), which were previously calibrated against NIST Standard Reference Materials. In addition to $\partial^{13}C$, $\partial^{15}N$ signatures, the percentage of organic tissue as nitrogen (%$N_{org}$), and the percentage of tissue as carbon (total: %$C_T$ and organic: %$C_{org}$) were obtained for each sample.

### 2.2 Species Identification

Rhodoliths were checked under a dissecting microscope for the presence of epi- and endophytes or other adhering foreign matter. Clean fragments were ground to a fine powder with a sterile mortar and pestel, and DNA was extracted using the Qiagen DNeasy Blood and Tissue Kit (Qiagen, Crawley, UK), following the modifications of the manufacturers' protocol given in Broom et al. (2008). PCR amplifications and sequencing of the *rbc*L gene were performed according to the methods given in Heesch et al. (2016), using the primers F57 and R1150 (Freshwater and Rueness, 1994). Sequences were deposited in ENA/GenBank (see Table S1 for accession numbers).

## 2.3 pH Drift Experiments

pH drift experiments were conducted for two Irish species, *Phymatolithon calcareum* (Pallas) W.H. Adey & D. L. McKibbin ex Woelkering & L. M. Irvine and *P. lusitanicum* (P. Crouan & H. Crouan) Woelkerling & L. M. Irvine, the Canary Islands samples, and the samples from Akia Penninsula, Greenland. These drift experiments can be used to determine if an efficient CCM is present, because if the seawater pH in the sealed incubation becomes greater than 9.0, no more $CO_2$ is present, and it can be assumed that $HCO_3^-$ is taken up during photosynthesis (Maberly, 1990). The incubations were conducted during a 24 – 70 hour light exposure with 35 µmol photons $m^{-2}$ $s^{-1}$ at 15°C, 20°C or 4°C (for the Irish, Canary Islands, and Greenland samples, respectively) in natural seawater (34 psu) in 200 - 300 ml glass or plastic jars, depending on the size of the specimens. The duration of the light exposure depended on the specimens. The Greenland specimens were incubated longer due to their slower metabolic rates. The seawater was vacuum filtered using 0.22 µm Durapore membrane filters (Merck Millipore, Darmstadt, Germany). The jars contained stir bars and were placed on a multi-position stir plate. The rhodoliths were held on custom-made stands inside the jars to allow space for the stir bars below. The start and finish pH values (NBS) were recorded using a Sentix 51 pH electrode with an integrated temperature probe connected to a WTW 3110 pH meter (Weilheim, Germany). The seawater used during the incubation was left open to the ambient air for re-equilibration to make sure the change in pH was due to the metabolism of the algae. Oxygen production rates were measured either simultaneously or in separate 1-hour incubations with oxygen sensor spots (OSXP5) glued to the inside of the glass jars (Pyroscience, Aachen, Germany). Four - five individuals of each species were measured.

### 2.3 Data Analysis & Statistics

In order to standardize the data used for the environmental parameters at each site, we obtained surface temperature, salinity, pH, total dissolved inorganic carbon (DIC) and total alkalinity (TA) data from locations closest to our sampling sites compiled on the Ocean Data View website (http://odv.awi.de/en/data/ocean/global-alkanity-tco2/) for global alkalinity and total dissolved carbon estimates from Goyet et al. (2000). Only surface data were used from the data set. For quality control, these data were compared to data from long-term monitoring stations at our sampling sites when available. From these data, the remaining parameters of the seawater carbonate system were calculated using the R package seacarb in RStudio (version 1.0.143).

In order to investigate specific relationships between the $\partial^{13}C$ signatures and environmental variables, a multiple regression analysis was conducted for $\delta^{13}C_{org}$ using DIC, temperature, latitude, $\%N_{org}$, $\partial^{15}N_{org}$, $\%C_T$, $\%C_{org}$, $\partial^{13}C_T$, salinity, and pH as regression factors. A relative importance test was conducted to determine the relative importance of each regression factor using the relaimpo package in R. The function calc.relimp was used to calculate relative importance metrics for the linear model, and the function boot.relimp was used to calculate bootstrap confidence intervals for the relative importance of each regression factor using the pratt method.

A principle component analysis (PCA) was applied to the data to identify patterns between physiological characteristics, species, collection site, and environmental variables. Environmental variables that had correlation coefficients greater than 0.95 were not both included in the analysis to avoid multicollinearity. A skewness transformation was applied, and the data were centered and scaled prior to PCA analysis. The PCA was conducted using the prcomp function in the R stats package and the results were visualized using the ggbiplot function in the R package ggplot2.

The effect of species on $\partial^{13}C$ signatures was tested using a multivariate analysis of variance with species as the independent variables and $\partial^{13}C_{org}$ or $\partial^{13}C_T$ as response variables. Tukeys HSD tests were conducted to determine which species differed from each other.

The effect of depth on $\partial^{13}C_T$, $\partial^{13}C_{org}$, $\partial^{15}N$, %$N_{org}$, %$C_{org}$ and %$C_T$ was analysed for the rhodolith samples from Mosselbukta using a multivariate analysis of variance. The data were checked for normality and homogeneity of variance using Shapiro-Wilk and Bartlett's tests, respectively. Significant differences between depths for each dependent variable were tested using Tukey's HSD tests.

# 3 Results

The collection site, species name, and accession numbers of the identified specimens are summarized in Table S1. The rhodoliths from Greenland were collected from two sites, Kobbel Fjord (64.14, -51.59) and Akia Penninsula (64.19, -51.91). The samples were identified as a mixture of two closely related species, *L. glaciale* Kjellman (from Akia and Kobbel Fjords) and a second entity from Kobbel Fjord, which also occurred in Oslo Fjord. This specimen was most closely related to *L. erinaceum* Melbourne & J.Brodie, a species recently described and reported from the UK (Melbourne et al., 2017). The specimens from western Ireland were a mixture of three species: *Phymatolithon lusitanicum* (V. Peña), *P. calcareum* (Pallas) W.H.Adey & D.L.McKibbin, and *Lithophyllum incrustans* Philippi. *P. lusticanicum* was collected from Carraroe, while the other two species were collected from Mannin Bay. The samples from Brest were identified as a mixture of *L. corallioides* (P.L.Crouan & H.M.Crouan) P.L.Crouan & H.M.Crouan and a closely related, undefined species. The Gran Canaria rhodoliths were tentatively identified as *Lithothamnion* sp., although both *Lithothamnion* and *Phymatolithon* sp. 2 (Pardo et al., 2014) are present off the eastern coast of Gran Canaria and the descriptions of these species are still in flux (Viviana Peña pers. comm.). The rhodoliths collected from Mosselbukta were identified as *L. glaciale* by Teichert et al. (2014). Our molecular analysis confirmed that these samples were dominated by *L. glaciale*, but we also identified a specimen of *L. lemoineae* Adey from our sub-sampling of this collection.

The $\delta^{13}C_T$ signatures of rhodoliths analysed in this study ranged from -11.9 to -0.89 and were generally higher than the $\delta^{13}C_{org}$ signatures, which ranged from -25.7 to -2.8 (Figure 1). Both *Phymatolithon* species collected from Ireland had significantly higher $\partial^{13}C_T$ signatures than *Lithothamnion glaciale*, and *P. calcareum* had significantly higher $\partial^{13}C_T$ than species in all other genera. Differences were less pronounced for $\partial^{13}C_{org}$, where *P. lusitanicum* had higher signatures than *L. glaciale* and *L. incrustans*.

The PCA analysis explained 79% of the variance within the first three components (Figure 2). The first axis (PC1) separated the cold-temperate/Arctic samples from the temperate and sub-tropical samples based on higher [DIC] and lower temperature, pH, [$CO_3^{2-}$] and salinity. The second axis (PC2) separated the samples based on $\partial^{15}N$ and $\partial^{13}C$. The Oslo, Brest and Irish *Lithophyllum* samples had relatively high $\partial^{15}N_{org}$ ratios compared to the others, while samples from Gran Canaria and the Irish *Phymatolithon* spp. had the highest $\partial^{13}C$ ratios. The Greenlandic specimens were strongly separated from all other groups due to the site having the lowest pH and [$CO_3^{2-}$]. The three species collected from Ireland were distinguished

based on $\partial^{13}C$ and $\partial^{15}N$ signatures. The $\partial^{13}C$ signatures (both organic and inorganic) increased in the order *L. incrustans* to *P. crispatum* to *P. lusitanicum*, and the $\partial^{15}N$ signatures showed the exact opposite trend.

Multiple regression analysis of only the *Lithothamnion* spp. showed a significant correlation between $\partial^{13}C_{org}$ and DIC, $\partial^{13}C_T$,
$\partial^{15}N_{org}$, %$C_{org}$, and salinity (Figure 3). The proportion of variance explained by the regression model was 94.4%. The total and organic fraction $\partial^{13}C$ signatures were most strongly correlated, and this correlation accounted for 86% of the $R^2$ in the model. Figure 4a shows the relationship between $\partial^{13}C_{org}$ and $\partial^{13}C_T$ by species (the dominant species at each site). All species showed a strong linear relationship. Figure 4b shows the relationship between $\partial^{13}C_{org}$ and % organic carbon. All species showed an increasing $\partial^{13}C_{org}$ with increasing organic carbon content, with the exception of *L. corallioides* and *L. glaciale*.
Temperature, latitude and salinity were the environmental variables with the strongest contributions to the model (32, 23 and 22%, respectively, Figure 5a). The proportion of variance explained by the regression model using $\partial^{13}C_T$ as a response variable was 93.8%, and DIC was the most important environmental factor (16.5% of the $R^2$; Figure 5b). Closer examination of the relationship between carbonate content and $\partial^{13}C_{org}$ signatures showed that within species, there was a negative linear relationship between carbonate content and $\partial^{13}C_{org}$ signatures (Figure 6).

Depth had a significant effect on $\partial^{13}C_T$ (F=4.923, p=0.02271), %$N_{org}$ (F=5.4079, p=0.01704), %$C_{org}$ (F=3.809, p=0.0459), and %$C_T$ (F=9.99, p=0.0017) of the Mosselbukta rhodoliths (Figure 7). The $\partial^{13}C$ signatures (total and organic) of rhodoliths collected at 11 m were significantly lower than those collected at 25 and 40 m, while the %N and %$C_T$ were significantly higher in the rhodoliths collected at 11 m compared to 25 and 40 m. The rhodoliths collected from 40 m had significantly
lower %$C_{org}$ than the rhodoliths collected at 11 and 25 m.

pH drift experiments showed that the seawater pH actually decreased after a 24 hour light incubation for both *Phymatolithon calcareum* and *P. lusitanicum* (Figure 9). However, these samples were small, and it is possible that the incubation period was not long enough to detect a significant change in pH. In comparison, the Canary Islands samples elevated the seawater
pH up to 9.07. The Greenland samples also increased the seawater pH up to 9.7, but the seawater pH did not return to ambient levels after being exposed to the atmosphere. After 5 days, the seawater pH was lower than when the rhodoliths were present, but still higher than the starting pH value. The Greenland samples produced high amounts of dissolved organic carbon during the incubation period (data not shown), which could have strongly affected the pH compensation point, and suggests that pH drift experiments for these specimens are not reliable methods for determining the pH compensation point.

**4 Discussion**

Our results demonstrate that $\partial^{13}C$ signatures in rhodoliths are highly variable, and that this variability is closely related to environmental factors, particularly temperature and seawater chemistry. However, $\partial^{13}C$ signatures are also species dependent, since two different species collected from the same site (*L. incrustans* and *P. calcareum*) showed significant differences in $\partial^{13}C$ signatures. In general, all the rhodoliths collected in this study had $\partial^{13}C_{org}$ signatures greater than -30,
suggesting that none of them are relying solely on diffusive $CO_2$ uptake. The mean $\partial^{13}C_{org}$ signatures for most species were close to -10, suggesting that most of the rhodoliths examined in this study have relatively efficient CCMs. Both *Phymatolithon* spp. consistently had $\partial^{13}C_{org}$ signatures greater than -10, suggesting that these species primarily take up $HCO_3^-$ directly. The pH drift experiments also support the hypothesis that most rhodoliths investigated have an active CCM involving $HCO_3^-$ uptake, since several individuals from the highest and lowest latitude investigated had pH compensation
points above 9.0. There was high variability in pH compensation points due to the variability in size and shape of the rhodoliths used for the incubations. For the Canary Islands samples, most individuals were similar in size, but some were more or less solid/hollow than others.

The $\partial^{13}C_{org}$ signatures measured in our study are relatively high for red algae (Rhodophyta) in general, and are higher than other coralline algae collected from Brittany (Schaal et al., 2009, 2012). Red algae have been reported to have generally lower $\partial^{13}C$ signatures than the green (Chlorophyta) and brown algae (Phaeophyceae) (Hepburn et al., 2011; Moulin et al., 2011; Raven et al., 2002; Stepien, 2015). However most of the reported values supporting this trend are for fleshy macroalgae (Maberly et al., 1992; Marconi et al., 2011; Raven et al., 2002). Our data complement the findings by Stepien (2015), who analysed published $\partial^{13}C$ signatures of marine macrophytes and reported that calcifying algae had the highest $\partial^{13}C_{org}$ signatures compared to all other functional groups. Furthermore, the presence or absence of CCMs in red algae is apparently not a conserved trait, as families within this group are strongly variable with respect to the presence or absence of CCMs (Stepien, 2015). Our study demonstrates that northeast Atlantic rhodoliths contribute to this variability, as they deviate from the typical trend of red macroalgae lacking or having inefficient CCMs. The high $\partial^{13}C_{org}$ signatures measured in our study support previous work suggesting that crustose coralline algae (CCA) can directly take up $HCO_3^-$ for calcification and photosynthesis (Comeau et al., 2013; Hofmann et al., 2016). Because CCA take up $HCO_3^-$ for calcification (Comeau et al., 2013), the same transporters are likely used to supply inorganic carbon for photosynthesis, which would explain the high $\partial^{13}C_{org}$ signatures. The positive linear relationship between $\partial^{13}C_T$ and $\partial^{13}C_{org}$ across all specimens suggests that there is a strong use of respiratory $CO_2$ during calcification in northeast Atlantic rhodoliths (Lee and Carpenter, 2001).

The linear relationship between skeletal $\partial^{13}C$ signatures ($\partial^{13}C_T$) and DIC observed in our study supports the notion that these signatures can be used as a proxy for seawater DIC in long-lived coralline algae (Williams et al., 2011). Although the $\partial^{13}C_{org}$ signatures of rhodoliths from multiple genera collected in our study did not show a strong relationship to DIC, when only rhodoliths from a single genus (*Lithothamnion*) were considered, there was a strong linear relationship to DIC. The increasing trend in *Lithothamnion* spp. $\partial^{13}C$ signatures with decreasing DIC indicates that this genus may exhibit physiological plasticity in carbon concentrating mechanisms what could facilitate adaptation to changing seawater chemistry induced by ocean acidification. Cornwall et al. (2017) has recently shown that macroalgae with flexible CCMs whose $CO_2$ use increased with $CO_2$ concentration were more abundant at natural $CO_2$ seeps. Although the authors also found that obligate calcifiers were less abundant at natural $CO_2$ seeps, as other studies have also shown (Fabricius et al., 2015; Hall-Spencer et al., 2008), it my be possible that the physiological plasticity of rhodoliths observed in this study may nevertheless facilitate adaptation at a rate comparable to current global climate change.

The high $\partial^{15}N$ signatures in the samples from Bay of Brest and Oslo Fjord may be due to uptake of anthropogenic sources of nitrogen because of the close proximity of the collection sites to major cities and high agricultural activity. Sewage discharge has been shown to elevate $\partial^{15}N$ signatures of macroalgae (McClelland et al., 1997; McClelland and Valiela, 1998). These values are higher than those reported for the coralline alga *Corallina elongata* collected from northern Brittany (Schaal et al., 2009, 2012), where anthropogenic nitrogen inputs are lower than in the Bay of Brest, which is influenced by strong agricultural activity. From September 2015-2017, the maximum nitrate concentration in the Bay of Brest was 37 µM $NO_3^-$, compared to 8.4 µM in Roscoff (Service d'Observation en Milieu Litoral, INSU-CNRS, Roscoff). A comparison of fleshy macroalgae from the Brest Harbour and Batz Island, a pristine environment off the coast of Roscoff, showed that $\partial^{15}N$ signatures were enriched in macroalgae from Brest Harbour (Schaal et al., 2010). Therefore, the $\partial^{15}N$ signatures of rhodoliths also appear to be impacted by anthropogenic nitrogen inputs, like in other macroalgae.

The increase in $\partial^{13}C$ signatures we observed with depth was surprising, considering most studies have observed decreases in $\partial^{13}C$ signatures with increasing depth (Hepburn et al., 2011; Stepien, 2015). Light has been shown to be an important factor influencing $\partial^{13}C$ signatures in macroalgae (Cornwall et al., 2015; Murru and Sandgren, 2004; Raven et al., 2002; Stepien et

al., 2016). Considering that rhodoliths are low-light adapted subtidal algae, it is possible that other factors such as temperature, DIC or water velocity have a stronger influence on $\partial^{13}C$ signatures than light in the case of rhodoliths. In fact, there may be a relationship between $\partial^{13}C$ signatures and dissolved organic carbon (DOC) availability in rhodolith beds, since rhodolith food webs depend strongly on external inputs of organic matter (Grall et al. 2006, Gabara 2015), and the biogeochemical cycling within the rhodolith bed food web influences isotopic signatures. A comparison of $\partial^{13}C_{org}$ values measured in this study with concentrations of DOC collected along a similar latitudinal gradient (compiled by the GLODAPv2 Group, Key et al. 2015) shows a similar decreasing trend with latitude (Fig. S1), but any direct relationship is purely speculative at this time. Investigating the influence of DOC on rhodolith physiology and $\partial^{13}C$ signatures is a topic that should be investigated further. Alternatively, different rates of DIC recycling at different depths in Arctic rhodoliths could influence their stable isotope signatures. Our data suggest that calcification in northeast Atlantic rhodoliths is strongly influenced by respiratory $CO_2$, but there could additionally be recycling of DIC from precipitated carbonate material that is re-dissolved in Arctic rhodoliths. If more DIC from re-dissolved carbonate is recycled in deep Arctic rhodoliths than in shallow specimens, that could explain the higher $\partial^{13}C$ signatures we observed at 25 and 40 m compared to 11 m.

In conclusion, our results show that many northeast Atlantic rhodolith species, particularly those at lower latitudes, have carbon concentrating mechanisms that facilitate $HCO_3^-$ use for photosynthesis. This is an important adaptation for marine macroalgae, because $HCO_3^-$ is available at higher concentrations than $CO_2$ in seawater, and this becomes even more extreme with increasing temperature. The flexibility of CCMs in northeast Atlantic rhodoliths observed in our study may provide a key physiological mechanism for potential adaptation of rhodoliths to future global climate change.

## 5 Author Contributions

LCH designed and carried out the experiments and wrote the manuscript. SH provided molecular taxonomy analysis for species identification and edited the manuscript.

## 6 Conflict of Interest

The authors declare that they have no conflict of interest.

## 7 Data Availability

The data have been archived in PANGAEA and the assigned DOI will be added to the final version of the manuscript. The DNA sequences of samples used in this study will be deposited to GenBank and the assigned accession numbers will be published in the supplementary material.

## 8 Acknowledgements

The authors would like to thank the following individuals for collection of rhodolith samples used in this study: Max Schwanitz, Kate Schoenrock, Stein Fredriksen, Sophie Martin, Fernando Tuya and Joy Smith. This research was funded by the National Science Foundation Ocean Sciences International Postdoctoral Research Fellow program awarded to Dr. L. C. Hofmann (grant number 1521610): "Plasticity of inorganic carbon use in marine calcifying macroalgae across a latitudinal gradient and consequences of global change."

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

**10 Figure Captions**

Figure 1. The A) organic ($\partial^{13}C_{org}$) and B) total ($\partial^{13}C_T$) stable carbon isotope signatures of rhodoliths collected for this study grouped by genus (green=*Lithothamnion*, red = *Lithophyllum*, blue = *Phymatolithon* and species (Lc = *L. corallioides*, Le = *L. erinaceum*. Lg = *L. glaciale*, Lg2= *L. glaciale2*, Lsp = *Lithothamnion* sp., Li = *L. incrustans*, Pc = *P. calcareum*, Pl = *P. lusitanicum*.

Figure 2. The organic ($\partial^{13}C_{org}$) and total ($\partial^{13}C_T$) stable carbon isotope signatures of all *Lithothamnion* spp. collected for this study as a function of latitude, excluding the Mosselbukta samples collected deeper than 11 m.

Figure 3. Principle component analysis (PCA) of the response variables measured ($\partial^{13}C_{org}$, $\partial^{13}C_T$, $\partial^{15}N_{org}$, % $CaCO_3$, C:N ratio) and envinormental factors (DIC = total dissolved inorganic carbon, pH, Salinity, CO3 = $[CO_3^{2-}]$, Temp = temperature, °C, Long = longitude). Latitude, the remaining carbonate chemistry paratmers ($HCO_3^-$, $pCO_2$, $\Omega_{aragonite,}$) and %$C_{org}$ were not included to avoid multiple colinearity. The points are grouped by collection site (GRN = Greenland, OSLO = Oslo Fjord, BREST = Bay of Brest, GC = Gran Canaria, IR = Ireland, SPIT = Spitsbergen, MOSS = Mosselbukta) and labelled by species (Lc = *L. corallioides*, Le = *L. erinaceum*. Lg = *L. glaciale*, Lg2 = *L. glaciale2*, Lsp = Lithothamnion sp., Li = *L. incrustans*, Pc = *P. calcareum*, Pl = *P. lusitanicum*.

Figure 4. The linear relationships between $\partial^{13}C_{org}$ signatures of *Lithothamnion* spp. collected for this study and $\partial^{13}C_T$, organic carbon content, total dissolved inorganic carbon (DIC), $\partial^{15}N_{org}$, and salinity. Linear regression analysis showed significant correlation coefficients for these factors.

Figure 5. The relationship between $\partial^{13}C_{org}$ and A) $\partial^{13}C_T$ and B) percent organic carbon for each species at each site (Brest=Bay of Brest, GC = Gran Canaria, IR=Ireland, MOSS=Mosselbukta, OSLO= Oslo Fjord, SPIT = Spitsbergen).

Figure 6. A) The mean skeletal ($\partial^{13}C$ = symbol color) and organic ($\partial^{13}C_{org}$ = symbol size) stable carbon isotope signatures of *Lithothamnion* spp. mapped in relationship to surface ocean temperature, excluding the Mosselbukta samples collected deeper than 11 m. B) The skeletal stable carbon isotope signatures as a function of total dissolved inorganic carbon (DIC).

Figure 7. Relationship between carbonate content (% calcium carbonate) and $\partial^{13}C_{org}$ signatures for each species (separated by panels) and collection site (indicated by colour). Lc = *Lithothamnion corallioides*, Le = *L. erinaceum*. Lg = *L. glaciale*, Lg2= *L. glaciale2*, Lsp = Lithothamnion sp., Li = *Lithophyllum incrustans*, Pc = *Phymatolithon calcareum*, Pl = *P. lusitanicum*. Collection Sites (GRN = Greenland, OSLO = Oslo Fjord, BREST = Bay of Brest, GC = Gran Canaria, IR = Ireland, SPIT = Spitsbergen, MOSS = Mosselbukta)

Figure 8. Boxplots of the A) organic and B) total $\partial^{13}C$ signatures, C) $\partial^{15}N$ signatures, D) organic nitrogen content, E) organic carbon content, and F) total carbon content of *L. glaciale* collected from three depths at Mosselbukta. Asterisks indicate significant differences between depths.

Figure 9. The pH compensation points (maximum pH reached during pH drift experiment) for the rhodoliths from Greenland (*L. glaciale*), Gran Canaria (*Lithothamnion* sp.) and Ireland (*P. calcareum* and *P. lusitanicum*).

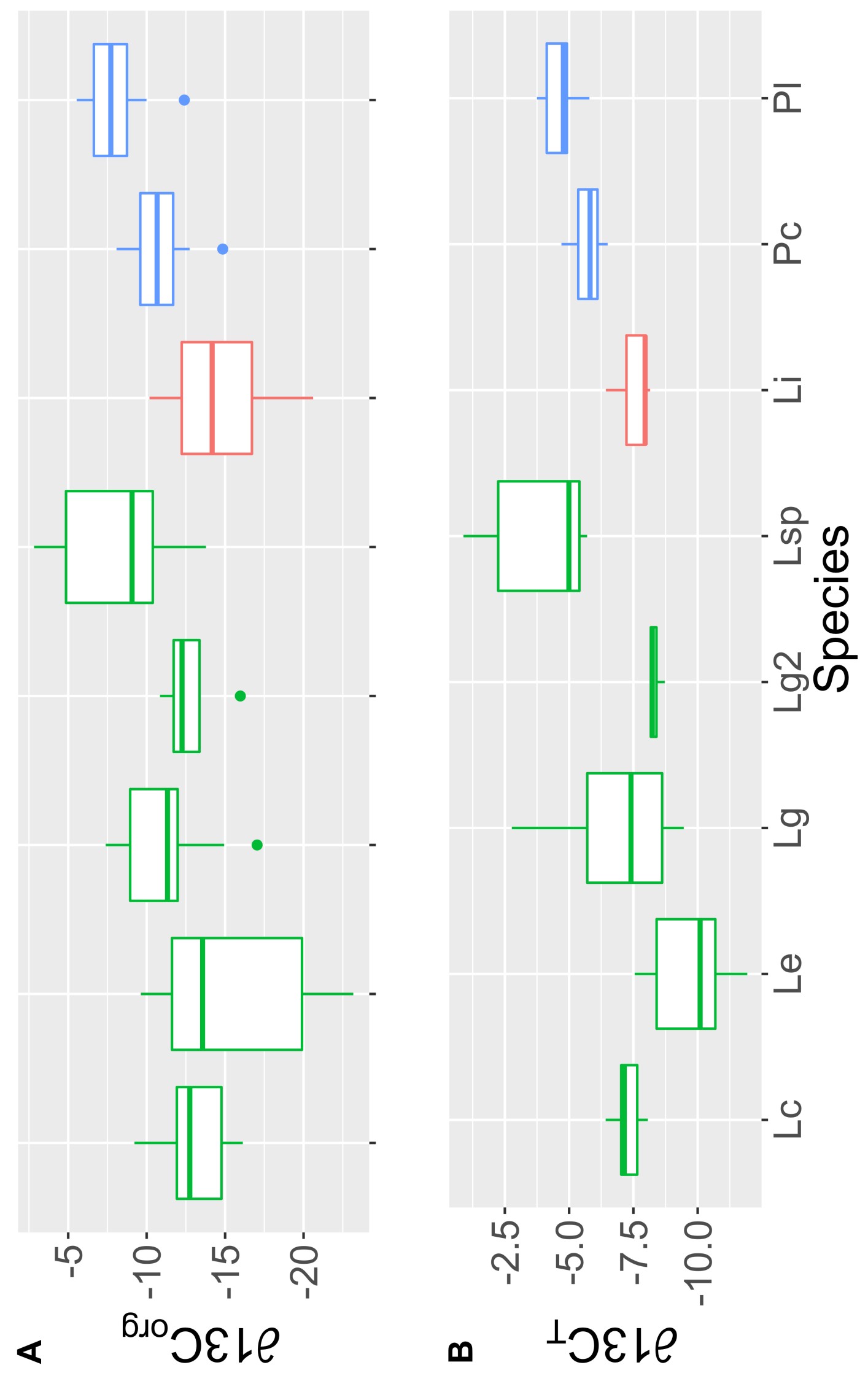

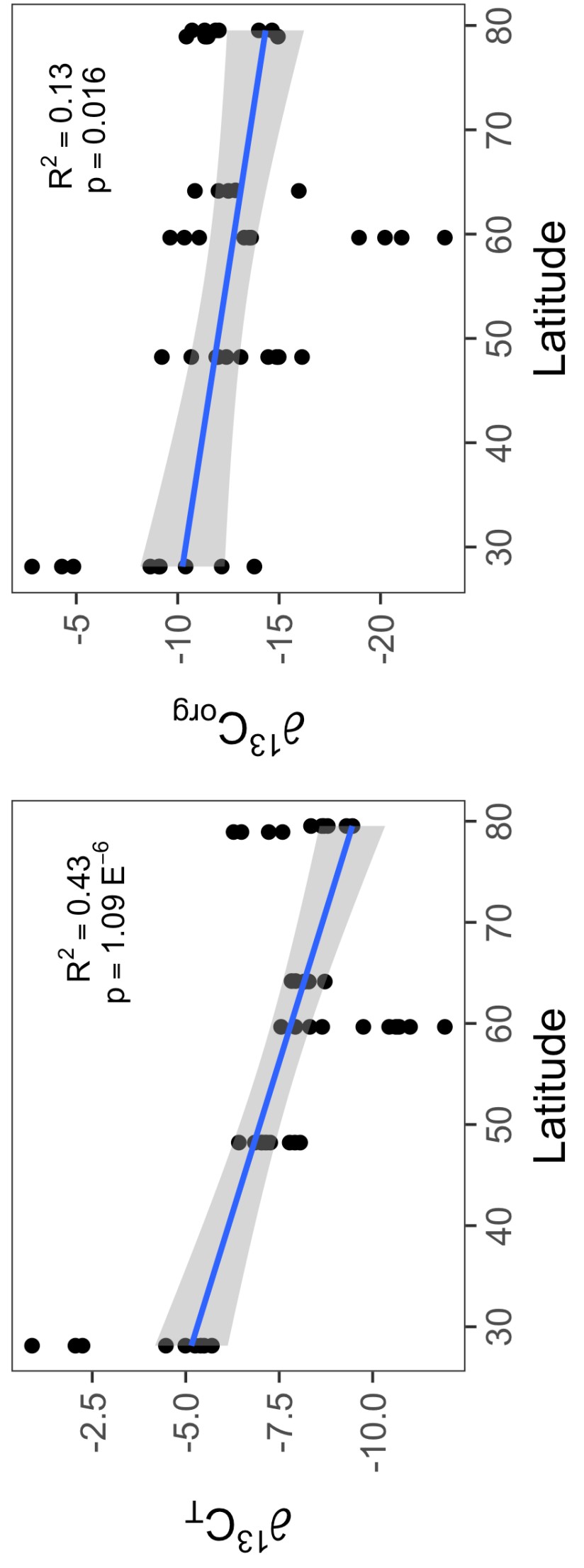

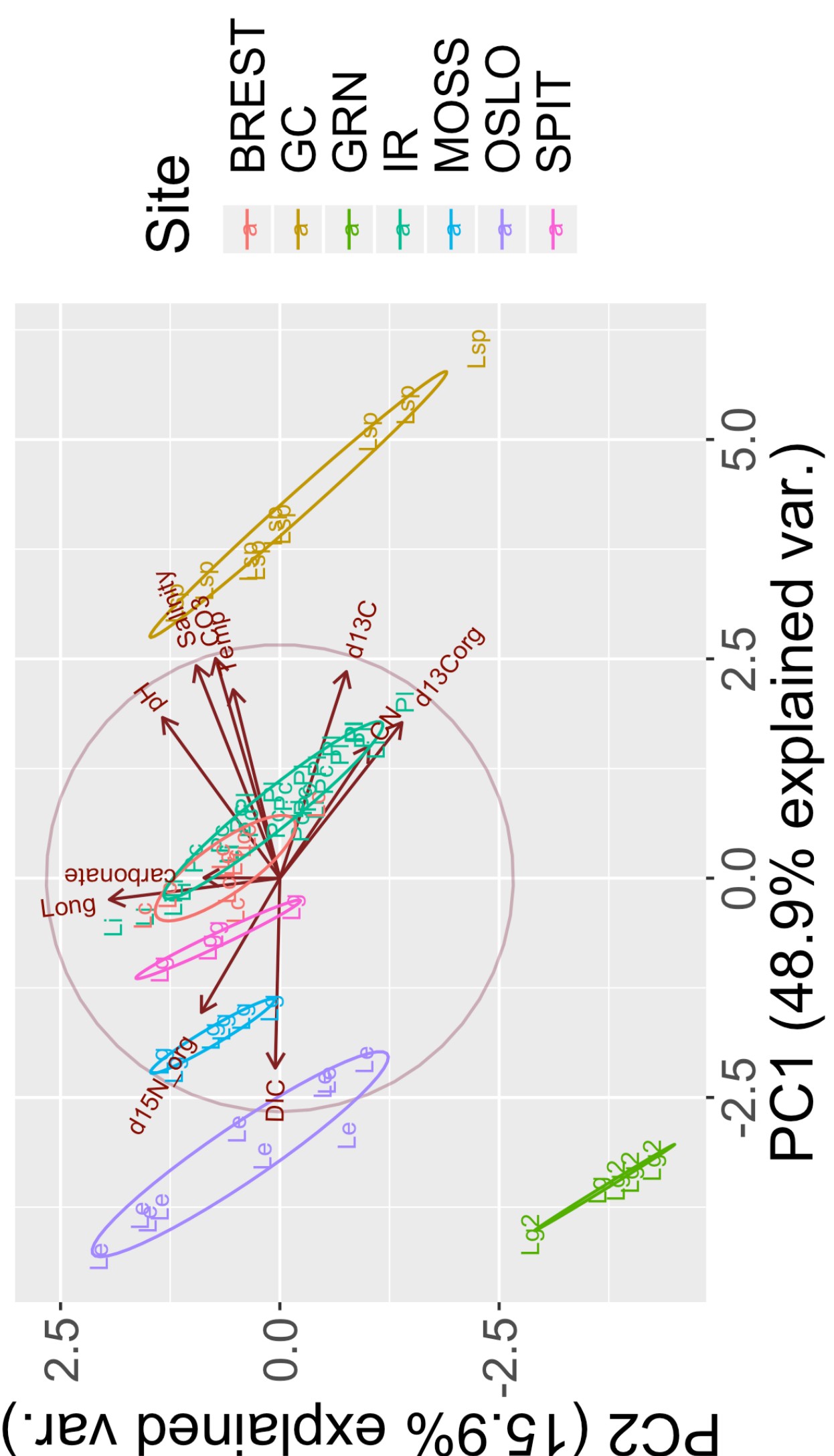

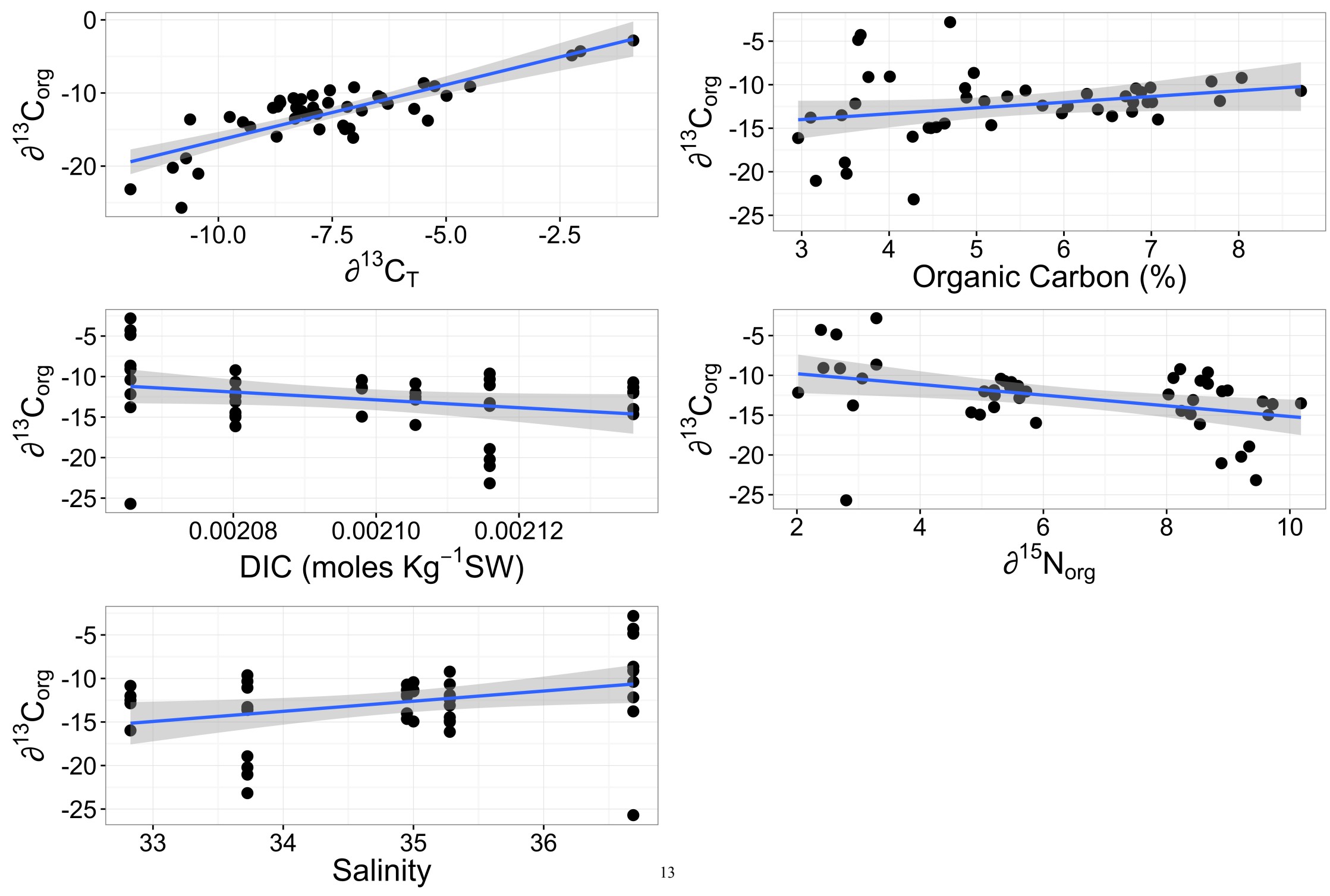

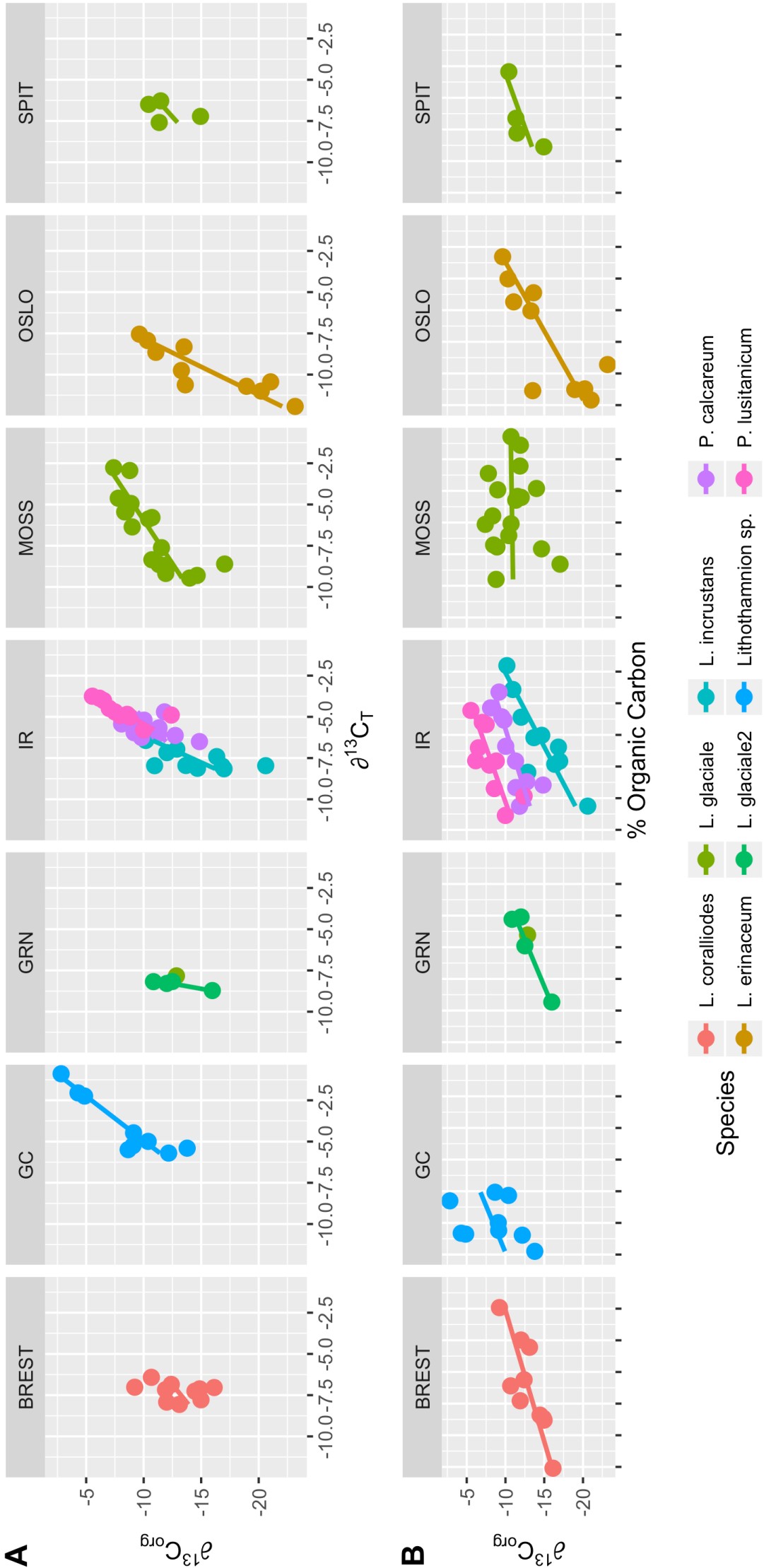

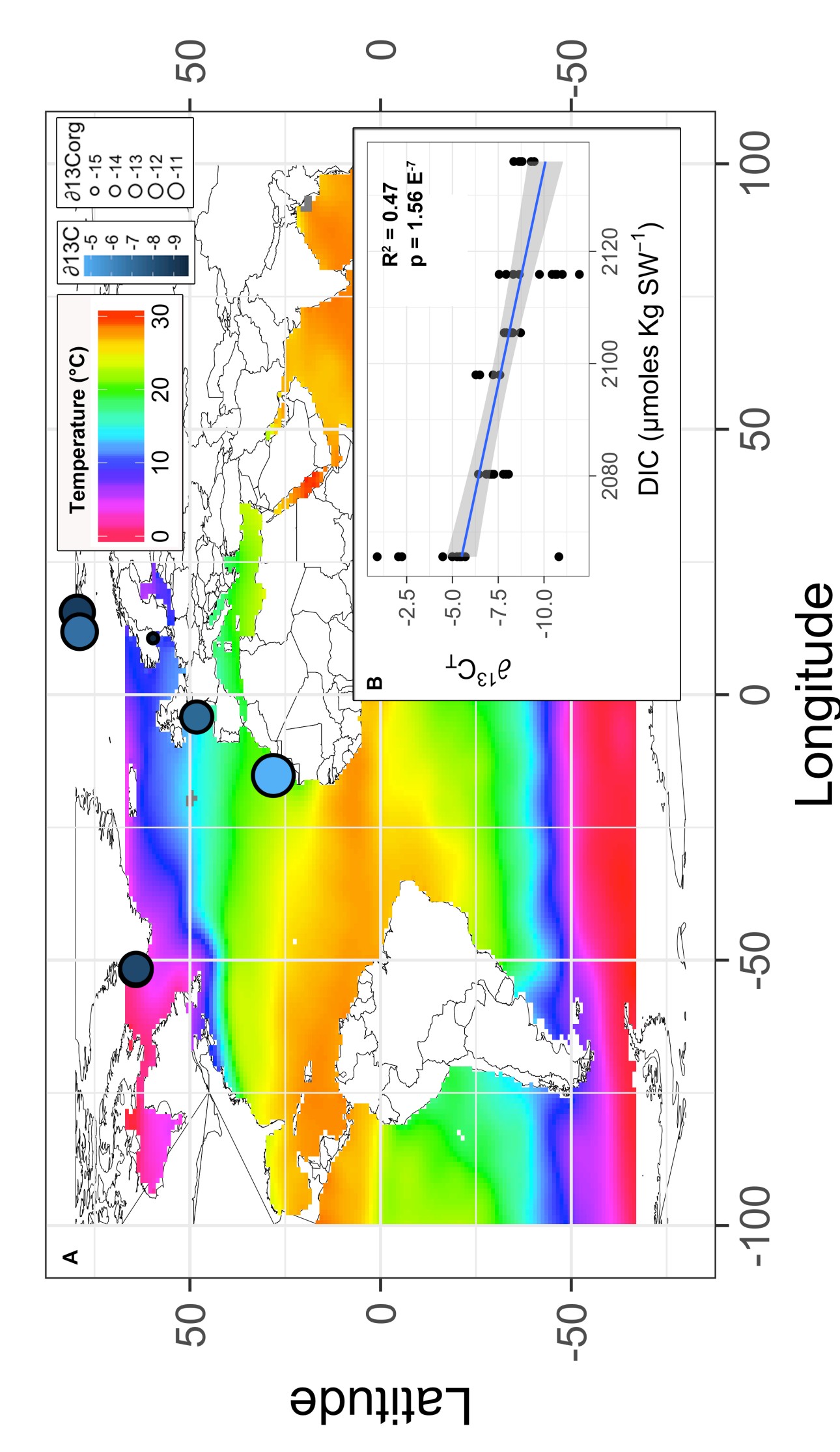

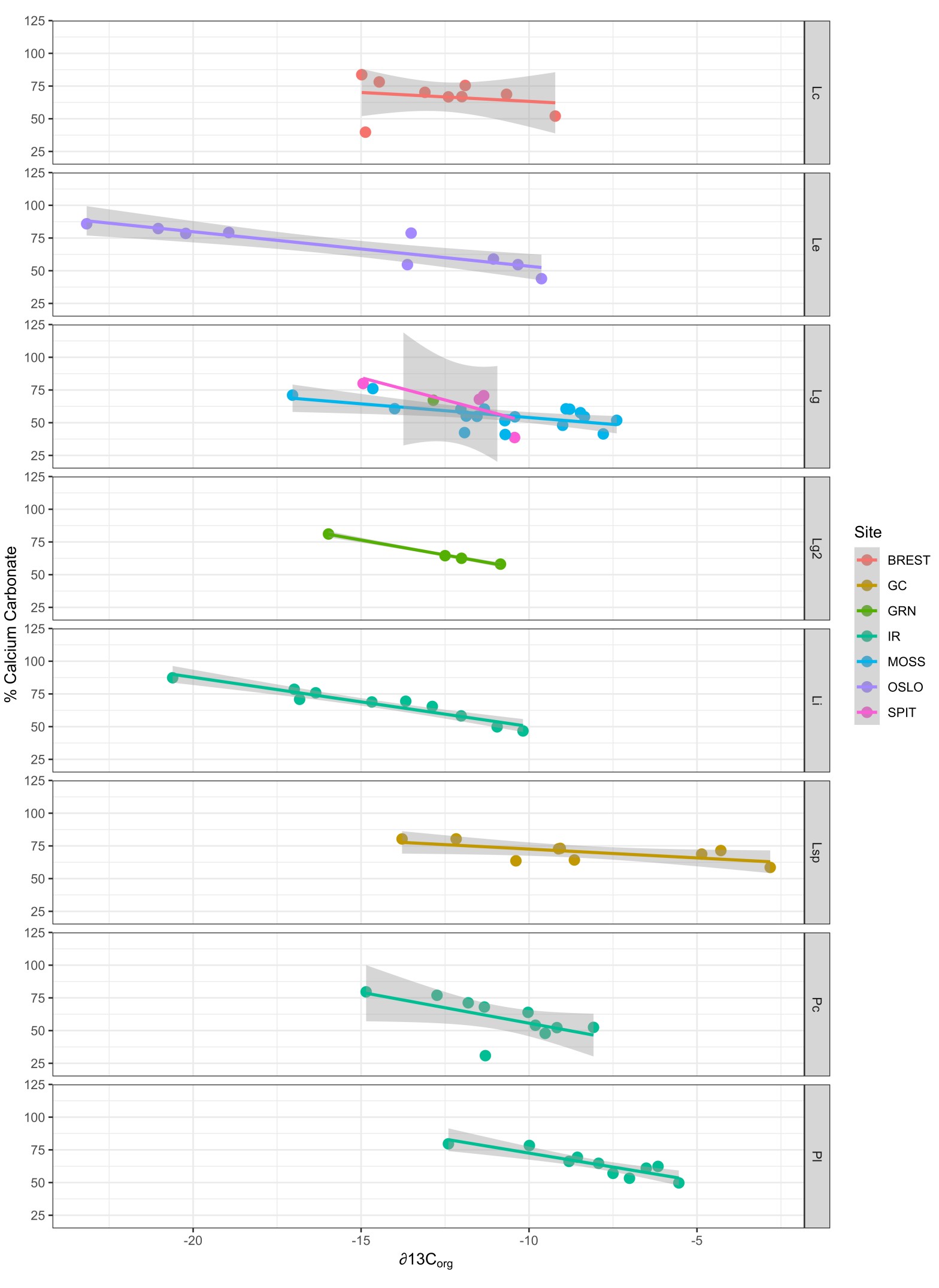

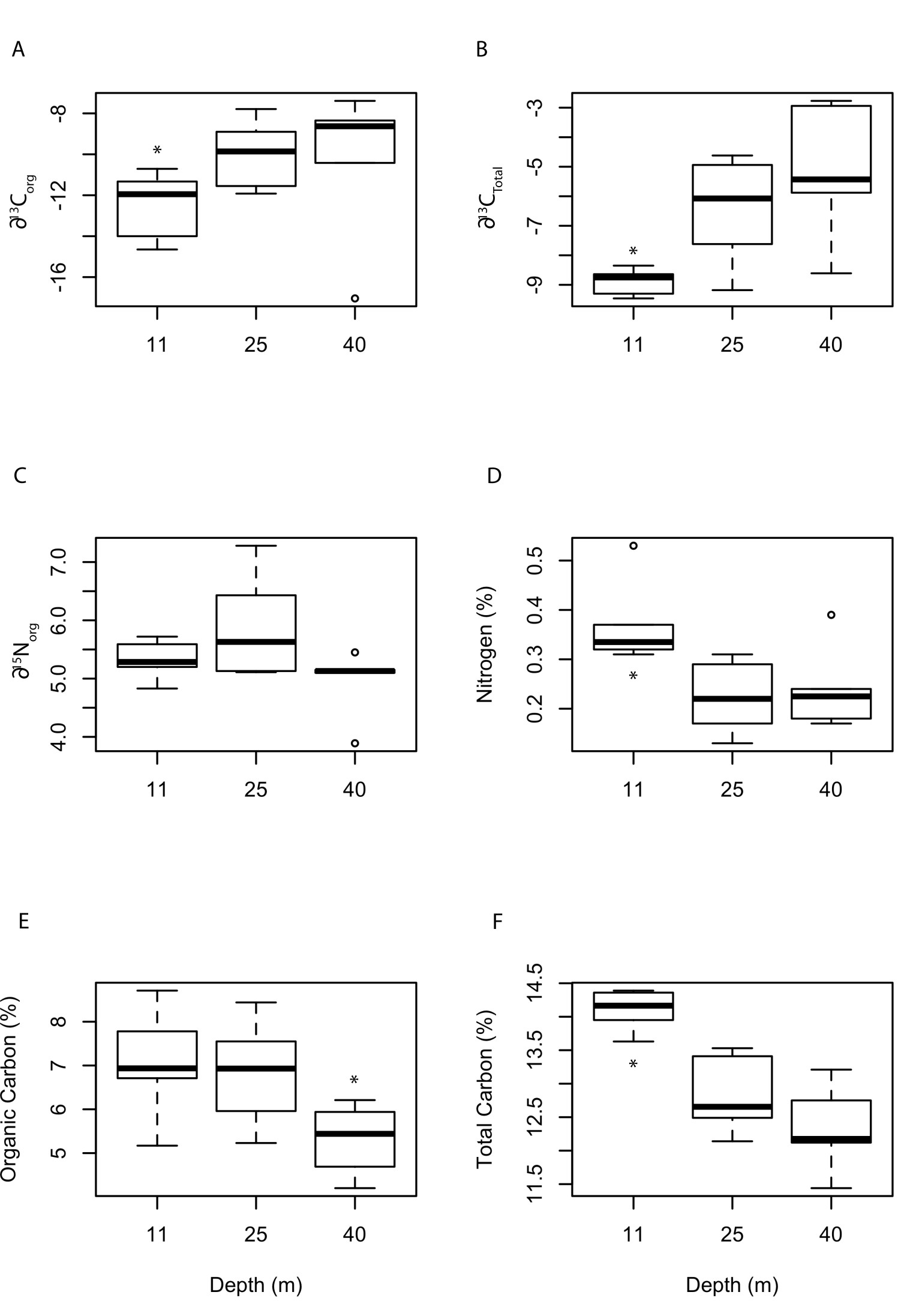

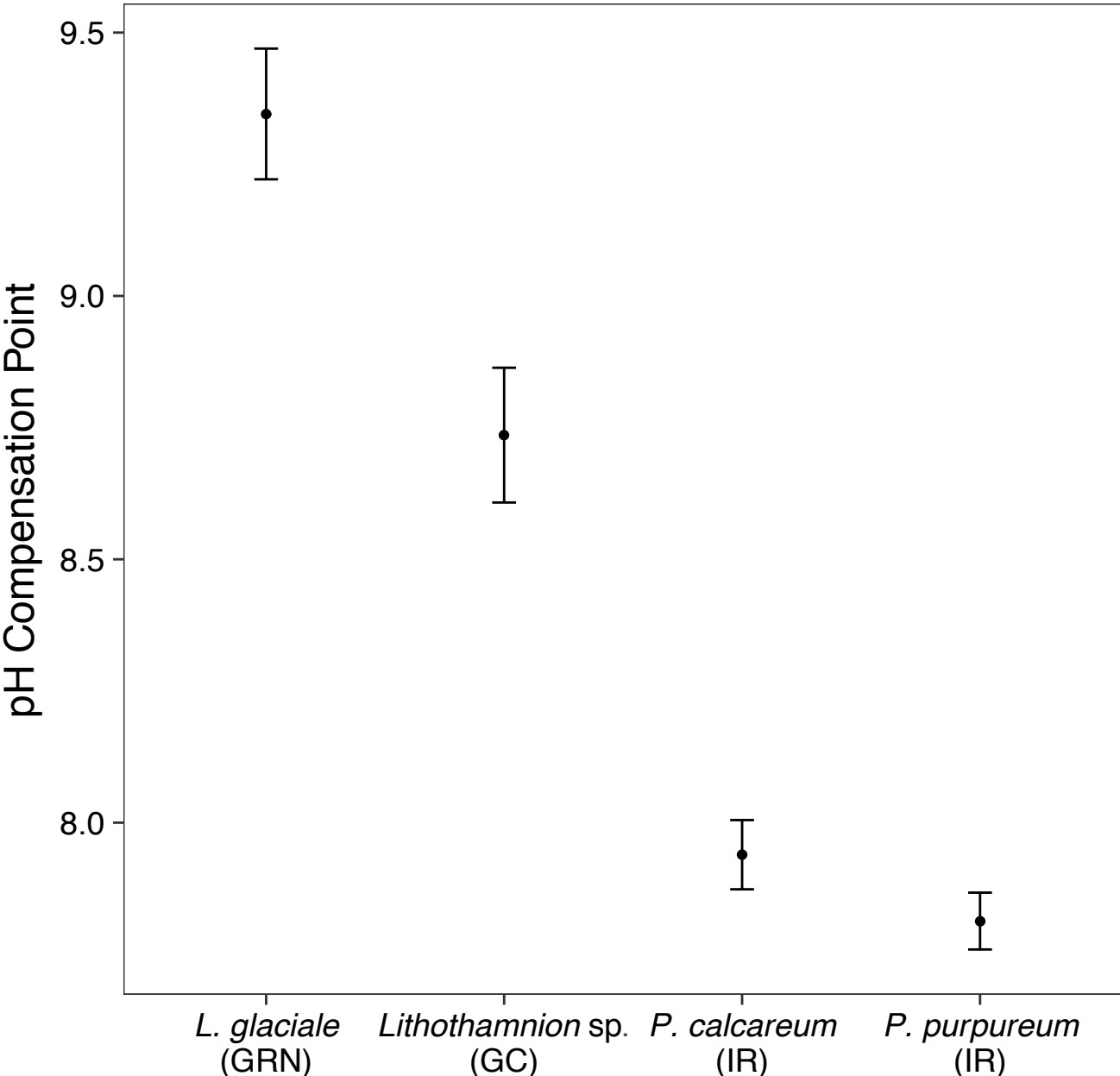