# Peer review of "Latitudinal trends in stable isotope signatures and carbon concentrating mechanisms of northeast Atlantic rhodoliths"

_Biogeosciences, 2017_

## Referee Comment (RC1) · Anonymous Referee #1 · 18 Oct 2017

The study by Hofmann and Heesch examines the d13C and related parameters of rhodolith species across a geographical gradient, with d13C being indicative of the proportion of HCO3- being directly taken up. I think this study is of good quality (I think) and useful, and has some really interesting findings that would be of use to phycologists. Below I detail some comments that could help the authors.

General comments I am somewhat confused with respects to the use of d13C total versus inorganic. In the discussion, the authors refer to past studies that ustlise inorganic d13C values. Can the authors confirm they did not bleach theses samples?

If so, I do not consider that they can compare the d13C total with inorganic values in past studies. In my specific comments I detail why. pH drift experiments and light experiments: There are no details regarding the light experiment (which first pops up in the discussion). This needs to be fully described and re-reviewed. Also, what was the water motion levels in the pH drift experiments? Did the authors measure pH over time to determine the ideal time to stop the drift experiments? The total pH drift does not seem high compared to past studies.

Specific comments Page 1, Line 17: I would be hesitant to claim that rhodoliths are obtaining all of their DIC from CO2 if their d13C value is ∼ -25. I would expect this to be lower (∼-29) – though I admit there is not much solid evidence that demonstrates whether the -29 "cut-off" is indeed due solely to CO2 uptake or CO2-facilitated uptake via exCA. Also, this is in complete contrast to the conclusions and discussion statements later on. Page 2, Line 5: My take is that d13C is an indicator of whether or direct HCO3- is being used. Similar d13C values could be obtained if the individual was relying solely on CO2 that was converted from exCA externally. The point here being that some in the community refer to exCA as a type of CCM. So this is more of a phrasing issue here. Page 2, line 26: Why was the total d13C measured, and not the inorganic compared to organic? I.e. bleaching the sample of organics before analysis. I think his would have yielded more interested results, as the total value could be influenced by both the organic content and by changes in the inorganic d13C and organic values. Page 3, lines 8 onwards: The type of pH buffers, instruments used to measure pH, TA and salinity are all needed here. Figure 4: Pretty interesting that Lc has such high organic d13C values. I have not seen that too often in CCA. Technical comment: could the key contain the species names rather than short-hands please. Figure 6: A map could be good as another panel here to compare the d13C versus the latitude of the sites. Figure 7: That is interesting that here depth is inversely related to d13C. So deeper specimens have less negative values. This is largely the opposite of what has been found with other species of macroalgae. I guess this is because other factors co-vary with depth that have stronger impacts here. I.er. organic carbon. Is

it possible that these specimens could be using DIC that has been modified by the calcification process, and because there is so little organics, perhaps they maintain a larger proportion of their DIC budget from DIC that was once carbonate that has been re-dissolved? Just a thought. Page 5, Line 37: I would suggest some clarity needs to be invoked around the term "use" here. It is likely that HCO3- is taken up directly by the organism. However, what is converted at the site of calcification in calcium carbonate might not be HCO3- at the nano-scale. In my opinion, HCO3- uptake takes place for the purposes of utilising as the DIC in calcification, but this is different to what might actually be precipitated. Again, this is a minor point, but it think the authors need to be clear about what is being said. Line 30, page 6: How did the authors determine what was new growth versus old here in the light experiments? Did they stain the organisms? Then how did they process the samples: One month could be too short to see adequate growth to get enough new inorganic material for d13C. I am not sure physiologically whether the organism would completely turn over the old organic tissue that was present at the start of the experiment too (as the authors allude to). Line 34, Page 6: I do not see the link physiologically with DOC and d13C. The authors need to explain this a little. A few other references that could be useful are Lee and Carpenter 2001 Chemical Geology, Cornwall et al 2013 Proc Roy Soc B, and Cornwall et al. 2014 Plos ONE, which all examine either d13C of organic or inorganic coralline algae (or both in the case of Cornwall et al 2013).

Technical comments Fgure5: Could the units of DIC be changed to umol per kg rather than moles. Figure7: Panel labelling is needed here. Some references in the text are missing from the bibliography. E.g. Page 6, line 27: Cornwall et al. 2015.

---

## Author Comment (AC1) · 13 Nov 2017

We would like to thank the anonymous referee for their constructive and helpful comments. Our detailed responses to the reviewers' comments are provided below. The revised version of the manuscript is attached, and we present our revised figures (Figs. 4, 5 and 7) based on the reviewer comments.

We would like to confirm that we did not bleach our samples for total $\partial$13C analysis. We acknowledge that some studies have bleached their samples to remove organic

[Figure]

=

material in the past. However, the specific citation referred to (Lee & Carpenter 2001) only used a weak bleach solution to remove surficial organic matter. Additionally, we did not find any evidence of treating samples for stable isotope analysis with bleach in studies using coralline algae as historical marine proxies (e.g. Hetzinger et al. 2009, Williams et al. 2011, Bougeois et al. 2015), which are the studies we compare our results to with respect to the total $\partial$13C analysis.

pH drift experiment

The pH drift experiments contained stir bars and were conducted on a magnetic stir plate with multiple magnets. We have added the statement (page 3, lines 8-9) "The jars contained stir bars and were placed on a multi-position stir plate. The rhodoliths were held on custom-made stands inside the jars to allow space for the stir bars below." We have removed the section on the carbon use model, considering that, as the reviewer mentioned, the pH change was much lower than would be expected, and it is possible that a 24 hour was not long enough to reach the pH compensation point. We address this on page 3, in lines 15-17 by stating ""pH drift experiments showed that the seawater pH actually decreased after a 24 hour light incubation for both Phymatolithon calcareum and P. purpureum (Figure 8). However, these samples were small, and it is possible that the incubation period was not long enough to detect a significant change in pH."

We have also repeated pH drift experiments for rhodolith samples from Greenland and the Canary Islands, the two extreme latitudes investigated in this experiment. We have changed Figure 8 to accommodate the new data, and add all necessary information to the methods, results and discussion accordingly as detailed below.

Page 3, Lines 9-16: The incubations were conducted during a 24 – 70 hour light cycle with 35 $\mu$mol photons m-2 s-1 at 15°C, 20°C or 4°C (for the Irish, Canary Islands, and Greenland samples, respectively) in natural seawater (34 psu) in 200 - 300 ml glass or plastic jars, depending on the size of the specimens. The duration of the light

=C2

=**[BGD]**
[Figure]

exposure depended on the specimens. The Greenland specimens were incubated longer due to their slower metabolic rates. The seawater was vacuum filtered using 0.22 $\mu$m Durapore membrane filters (Merck Millipore, Darmstadt, Germany). The jars contained stir bars and were placed on a multi-position stir plate. The rhodoliths were held on custom-made stands inside the jars to allow space for the stir bars below. The start and finish pH values (NBS) were recorded using a Sentix 51 pH electrode with an integrated temperature probe connected to a WTW 3110 pH meter (Weilheim, Germany). The seawater used during the incubation was left open to the ambient air for re-equilibration to make sure the change in pH was due to the metabolism of the algae.

Page 5, Lines 15-20 "pH drift experiments showed that the seawater pH actually decreased after a 24 hour light incubation for both Phymatolithon calcareum and P. purpureum (Figure 8). However, these samples were small, and it is possible that the incubation period was not long enough to detect a significant change in pH. In comparison, the Canary Islands samples elevated the seawater pH up to 9.07. The Greenland samples also increased the seawater pH up to 9.7, but the seawater pH did not return to ambient levels after being exposed to the atmosphere. After 5 days, the seawater pH was lower than when the rhodoliths were present, but still higher than the starting pH value. The Greenland samples produced high amounts of dissolved organic carbon during the incubation period (data not shown), which could have strongly affected the pH compensation point, and suggests that pH drift experiments for these specimens are not reliable methods for determining the pH compensation point."

Page 5, Lines 29-31: "The pH drift experiments also support the hypothesis that most rhodoliths investigated have an active CCM involving $HCO_3^-$ uptake, since several individuals from the highest and lowest latitude investigated had pH compensation points above 9.0."

Figure 8. The A) pH compensation points (maximum pH reached during pH drift experiment) and B) delta pH (final – initial) for the samples from Ireland (Phymatolithon spp.

from Mannin Bay and Carraroe) the Canary Islands, and Greenland (Akia Penninsula).

Light experiment

As this experiment was not a major aspect of the study, and we did not separate new growth from old growth, we have removed any reference to it from the manuscript (see also comment below).

Page 1, line 17: Thank you for pointing this out. The referee is correct that this statement does not support our conclusions and is not accurate according to our data. We have revised the statement as follows:

"We observed a decreasing trend in $\delta$13CT signatures with increasing latitude and temperature, while $\delta$13Corg signatures were only significantly correlated to DIC. These data suggest that high latitude rhodoliths rely more on $CO_2$ as an inorganic carbon source, while low latitudes rhodoliths likely take up $HCO_3^-$ directly, but none of our specimens had $\partial$13Corg signatures less than -30, suggesting that none of them relied solely on diffusive $CO_2$ uptake."

Page 2, line 5: We have rephrased the statement to: "Therefore, the ratio of stable carbon isotopes ($\partial$13C) in macroalgal tissue can be used as an indicator of whether or not $HCO_3^-$ is being used (Raven et al., 2002)"

Page 2, line 26: As mentioned above, this made our results more comparable to studies that investigated using coralline algae stable isotopes as marine proxies.

Page 3, line 8 onewards: More specific information on the experimental materials has been added. Because we removed the section on the carbon use model, we also removed the statement on total alkalinity analysis. The text now reads "The incubations were conducted during a 24 – 70 hour light cycle with 35 $\mu$mol photons m-2 s-1 at 15°C, 20°C or 4°C (for the Irish, Canary Islands, and Greenland samples, respectively) in natural seawater (34 psu) in 200 - 300 ml glass or plastic jars, depending on the size of the specimens. The duration of the light exposure depended on the specimens. The Greenland specimens were incubated longer due to their slower metabolic rates. The seawater was vacuum filtered using 0.22 $\mu$m Durapore membrane filters (Merck Millipore, Darmstadt, Germany). The jars contained stir bars and were placed on a multi-position stir plate. The rhodoliths were held on custom-made stands inside the jars to allow space for the stir bars below. The start and finish pH values (NBS) were recorded using a Sentix 51 pH electrode with an integrated temperature probe connected to a WTW 3110 pH meter (Weilheim, Germany). The seawater used during the incubation was left open to the ambient air for re-equilibration to make sure the change in pH was due to the metabolism of the algae."

Fig 4 – the legend now includes the abbreviated species names

Fig 5 – the units have been changed to $\mu$mol Kg SW-1 and we have added a map of surface ocean temperature with the mean d13C and d13Corg of Lithothamnion spp. plotted at their respective collection sites.

Fig 7 – panel labeling has been added

Page 5, Line 37: we have changed the wording to "directly take up HCO3-"

Page 6:

Line 30 - The reference to this experiment has been removed, since we have also removed the data from the supplementary, and because we did not separate new growth from old growth.

Line 34 - The link between DOC & $\partial$13C has been more clearly explained by this statement: "In fact, there may be a relationship between $\partial$13C signatures and dissolved organic carbon (DOC) availability in rhodolith beds, since rhodolith food webs depend strongly on external inputs of organic matter (Grall et al. 2006, Gabara 2015), and the biogeochemical cycling within the rhodolith bed food web influences isotopic signatures."

Cornwall et al. 2015 has been added to the reference list

Please also note the supplement to this comment:
https://www.biogeosciences-discuss.net/bg-2017-399/bg-2017-399-AC1-supplement.pdf

———————————————————

[Figure]

**Fig. 1.**

[Figure]

**Fig. 2.**

[Figure]

**Fig. 3.**

**Supplement:**

**Latitudinal trends in stable isotope signatures and carbon concentrating mechanisms of northeast Atlantic rhodoliths**

Laurie C. Hofmann[1], Svenja Heesch[2]

[1]Microsensor Group, Max Planck Institute for Marine Microbiology, Bremen, 28359, Germany
5  [2]Centre National de Recherche Scientifique, UMR 8227, Station Biologique de Roscoff, Roscoff, 29680, France

*Correspondence to*: Laurie C. Hofmann (lhofmann@mpi-bremen.de)

**Abstract.** Rhodoliths are free-living calcifying red algae that form extensive beds in shallow marine benthic environments (< 250 m), which provide important habitats and nurseries for marine organisms and contribute to carbonate sediment accumulation. There is growing concern that these organisms are sensitive to global climate change, yet little is known about 10 their physiology. Considering their broad distribution along most continental coastlines, their potential sensitivity to global change could have important consequences for the productivity and diversity of benthic coastal environments. The goal of this study was to determine the plasticity of dissolved inorganic carbon (DIC) uptake mechanisms of rhodoliths along a latitudinal gradient in the Northeast (NE) Atlantic using natural stable isotope signatures. The $\delta^{13}C$ signature of macroalgae can be used to provide an indication of the preferred inorganic carbon source ($CO_2$ vs. $HCO_3^-$). Here we present the total 15 ($\delta^{13}C_T$) and organic ($\delta^{13}C_{org}$) $\delta^{13}C$ signatures of NE Atlantic rhodoliths with respect to changing environmental conditions along a latitudinal gradient from the Canary Islands to Spitsbergen. The $\delta^{13}C_T$ signatures (-11.9 to -0.89) of rhodoliths analysed in this study were generally higher than the $\delta^{13}C_{org}$ signatures, which ranged from -25.7 to -2.8. We observed a decreasing trend in $\delta^{13}C_T$ signatures with increasing latitude and temperature, while $\delta^{13}C_{org}$ signatures were only significantly correlated to DIC. These data suggest that high latitude rhodoliths rely more on $CO_2$ as an inorganic carbon source, while 20 low latitudes rhodoliths likely take up $HCO_3^-$ directly, but none of our specimens had $\partial^{13}C_{org}$ signatures less than -30, suggesting that none of them relied solely on diffusive $CO_2$ uptake. However, depth also has a significant effect on both skeletal and organic $\delta^{13}C$ signatures, suggesting that both local and latitudinal trends influence the plasticity of rhodolith inorganic carbon acquisition and assimilation. Our results show that many species, particularly those at lower latitudes, have carbon concentrating mechanisms that facilitate $HCO_3^-$ use for photosynthesis. This is an important adaptation for marine 25 macroalgae, because $HCO_3^-$ is available at higher concentrations than $CO_2$ in seawater, and this becomes even more extreme with increasing temperature. The flexibility of CCMs in northeast Atlantic rhodoliths observed in our study may provide a key physiological mechanism for potential adaptation of rhodoliths to future global climate change.

Copyright statement: The authors have read and agree to the copyright terms outlined by the journal.

[revised manuscript text omitted]

Gabara, S. S.: Community structure and energy flow within rhodolith habitats at Santa Catalina Island, CA. Masters Thesis. San Jose State University, pp. 88, Available from: https://search.proquest.com/openview/c4cb6b5d922eec96c55037a1c87c0 bc7/1?pq-origsite=gscholar&cbl=18750&diss=y (Accessed 6 November 2017), 2015.

Giordano, M., Beardall, J. and Raven, J. a: CO2 concentrating mechanisms in algae: mechanisms, environmental
10  modulation, and evolution., Annu. Rev. Plant Biol., 56, 99–131, doi:10.1146/annurev.arplant.56.032604.144052, 2005.

Grall, J., Le Loc'h, F., Guyonnet, B. and Riera, P.: Community structure and food web based on stable isotopes ($\delta 15N$ and $\delta 13C$) analysis of a North Eastern Atlantic maerl bed, J. Exp. Mar. Bio. Ecol., 338(1), 1–15, doi:10.1016/J.JEMBE.2006.06.013, 2006.

[revised manuscript text omitted]

35  initial) for the samples from Ireland (*Phymatolithon* spp. from Mannin Bay and Carraroe) the Canary Islands, and Greenland (Akia Penninsula).

---

## Referee Comment (RC2) · Anonymous Referee #2 · 12 Jun 2018

The authors of this manuscript collected rhodoliths along a latitudinal gradient in the Northeast Atlantic and analyzed the stable carbon isotope signatures to indicate the carbon concentrating mechanisms. The study bears very useful information for potential adaptation of rhodliths to future global changes. I suggest publication of this manuscript after revisions.

General comments:

1. As the title of the manuscript is "Latitudinal trends in stable isotope signatures and carbon concentrating mechanisms of northeast Atlantic rhodoliths", one would expect to see what kind of latitudinal trends are discovered by the authors and what the mechanisms behind these trends are. In the results and discussion parts of this manuscript, these cores issues are somewhat not clearly targeted and/or not well presented/interpreted. For example, the authors better add a map showing location of the sampling sites and better move Figure S2 into the manuscript (it is the latitudinal trend!). I would suggest the authors either change the title of the manuscript or re-organize the results and discussion parts a little bit.

2. Materials and methods. The authors claimed that the samples are sent to UC Davis for stable isotope analysis. Since isotope signatures are vital to the discussion of this manuscript, the authors better give a brief introduction of the analytical methods or add some references about the analytical methods although the analysis might be done in prestige institutions. This way readers can trace the quality of the data. This is also necessary for other parameters like $\%N_{org}$, $\%C_T$ and $\%C_{org}$.

3. pH drift experiment and light exposure experiment. It is not so clear why the drift experiments can help to determine if an efficient CCM is present. As the incubation was done just over a 24hr light cycle. Is the incubation in a closed system resembling that in the real environment? If this incubation time is long enough for to yield robust results? As for the light exposure experiment, the authors did not describe how they were done in the "Materials and methods" part, and they did not explain why these light intensities were chosen for the experiment.

Minors:

Page 1 line 25: "CCMs" should be "carbon concentrating mechanisms" as CCMs were not defined before.

In the whole text, one would get confused by the mixing use of $\delta^{13}C$, $\partial^{13}C$ and $\mathbf{d}^{15}N$ for the isotope signatures!

Page 4 line 5: what the figures in the parentheses behind "Kobbel Ford" and "Akia Peninsula" mean?

---

## Author Comment (AC2) · 18 Jun 2018

The authors would like to express their gratitude to the second reviewer for their thoughtful and constructive comments. Our response to his/her comments are below.

General comments:

1. We have added a figure (Figure 2 - see attached) to show the direct relationship between d13C signatures and latitude for our samples. We also added a map showing our sites and the corresponding mean d13C signatures, based on suggestions from

the first reviewer (Fig. 6 - see attached).

2. We have added a description of the methods for d13C and d15N analysis used by the UC Davis stable isotope facility.

"The samples were analyzed for 13C and 15N isotopes using a PDZ Europa ANCA-GSL elemental analyzer interfaced to a PDZ Europa 20-20 isotope ratio mass spectrometer (Sercon Ltd., Cheshire, UK). Samples were combusted at 1000°C in a reactor packed with chromium oxide and silvered copper oxide. Following combustion, oxides were removed in a reduction reactor (reduced copper at 650°C). N2 and CO2 were separated on a Carbosieve GC column (65°C, 65 mL/min) before entering the IRMS. During analysis, samples were interspersed with several replicates of different laboratory standards (glutamic acid and peach leaves), which were previously calibrated against NIST Standard Reference Materials.

3. Based on similar comments from the first reviewer, our light exposure experiment has been removed from the manuscript and we have improved the description of the methods (see comments to first reviewer) and presentation of data from our pH drift experiments (Fig 9 - see attached). We have removed the section on the carbon use model, considering that, as the first reviewer mentioned, the pH change was much lower than would be expected, and it is possible that a 24 hour was not long enough to reach the pH compensation point. We will address this by stating "pH drift experiments showed that the seawater pH actually decreased after a 24 hour light incubation for both Phymatolithon calcareum and P. purpureum. However, these samples were small, and it is possible that the incubation period was not long enough to detect a significant change in pH."

Minors:

1. CCMs are now defined at the beginning of the abstract

2. We have edited the symbols so that $\partial$ is used throughout

3. The numbers behind the parenthesis represent lat/long coordinates

[Figure]

[Figure]

**Fig. 1.** Figure 2. The organic ($\partial$13Corg) and total ($\partial$13CT) stable carbon isotope signatures of all Lithothamnion spp. collected for this study as a function of latitude, excluding the Mossel-bukta samples colle

[Figure]

**Fig. 2.** Figure 6. A) The mean skeletal ($\partial$13C = symbol color) and organic ($\partial$13Corg = symbol size) stable carbon isotope signatures of Lithothamnion spp. mapped in relationship to surface ocean temperature, exc

[Figure]

**Fig. 3.** Figure 9. The pH compensation points (maximum pH reached during pH drift experiment) for the rhodoliths from Greenland (GRN), Gran Canaria (GC) and Ireland (IR).

---

## Author Response (AR2)

**Response to Reviewer 1**

We would like to thank the anonymous referee for their constructive and helpful comments. Our detailed responses to the reviewers' comments are provided below.

We would like to confirm that we did not bleach our samples for total $\partial$13C analysis. We acknowledge that some studies have bleached their samples to remove organic material in the past. However, the specific citation referred to (Lee & Carpenter 2001) only used a weak bleach solution to remove surficial organic matter. Additionally, we did not find any evidence of treating samples for stable isotope analysis with bleach in studies using coralline algae as historical marine proxies (e.g. Hetzinger et al. 2009, Williams et al. 2011, Bougeois et al. 2015), which are the studies we compare our results to with respect to the total $\partial$13C analysis.

pH drift experiment

The pH drift experiments contained stir bars and were conducted on a magnetic stir plate with multiple magnets. We have added the statement (page 3, lines 19-20) "The jars contained stir bars and were placed on a multi-position stir plate. The rhodoliths were held on custom-made stands inside the jars to allow space for the stir bars below." We have removed the section on the carbon use model, considering that, as the reviewer mentioned, the pH change was much lower than would be expected, and it is possible that a 24 hour was not long enough to reach the pH compensation point. We address this on page

5, in lines 21-24 by stating ""pH drift experiments showed that the seawater pH actually decreased after a 24 hour light incubation for both *Phymatolithon calcareum* and *P. lusitanicum* (Figure 9). However, these samples were small, and it is possible that the incubation period was not long enough to detect a significant change in pH."

We have also repeated pH drift experiments for rhodolith samples from Greenland and the Canary Islands, the two extreme latitudes investigated in this experiment. We have changed Figure 9 to accommodate the new data, and add all necessary information to the methods, results and discussion accordingly as detailed below.

Page 3, Lines 14-23: The incubations were conducted during a 24 – 70 hour light cycle with 35 µmol photons $m^{-2}$ $s^{-1}$ at 15°C, 20°C or 4°C (for the Irish, Canary Islands, and Greenland samples, respectively) in natural seawater (34 psu) in 200 -

300 ml glass or plastic jars, depending on the size of the specimens. The duration of the light exposure depended on the specimens. The Greenland specimens were incubated longer due to their slower metabolic rates. The seawater was vacuum filtered using 0.22 µm Durapore membrane filters (Merck Millipore, Darmstadt, Germany). The jars contained stir bars and were placed on a multi-position stir plate. The rhodoliths were held on custom-made stands inside the jars to allow space for the stir bars below. The start and finish pH values (NBS) were recorded using a Sentix 51 pH electrode with an integrated temperature probe connected to a WTW 3110 pH meter (Weilheim, Germany). The seawater used during the incubation was left open to the ambient air for re-equilibration to make sure the change in pH was due to the metabolism of the algae.

Page 5, Lines 21-29 "pH drift experiments showed that the seawater pH actually decreased after a 24 hour light incubation for both *Phymatolithon calcareum* and *P. lusitanicum* (Figure 8). However, these samples were small, and it is possible that the incubation period was not long enough to detect a significant change in pH. In comparison, the Canary Islands samples elevated the seawater pH up to 9.07.  The Greenland samples also increased the seawater pH up to 9.7, but the seawater pH did not return to ambient levels after being exposed to the atmosphere. After 5 days, the seawater pH was lower than when the rhodoliths were present, but still higher than the starting pH value. The Greenland samples produced high amounts of dissolved organic carbon during the incubation period (data not shown), which could have strongly affected the pH compensation point, and suggests that pH drift experiments for these specimens are not reliable methods for determining the pH compensation point."

Page 5, Lines 38-40: "The pH drift experiments also support the hypothesis that most rhodoliths investigated have an active CCM involving $HCO_3^-$ uptake, since several individuals from the highest and lowest latitude investigated had pH compensation points above 9.0."

Figure 9. The pH compensation points (maximum pH reached during pH drift experiment) for the rhodoliths from Greenland
(GRN), Gran Canaria (GC) and Ireland (IR).

Light experiment

As this experiment was not a major aspect of the study, and we did not separate new growth from old growth, we have removed any reference to it from the manuscript (see also comment below).

Page 1, line 17:

Thank you for pointing this out. The referee is correct that this statement does not support our conclusions and is not
accurate according to our data. We have revised the statement as follows:

"We observed a decreasing trend in $\delta^{13}C_T$ signatures with increasing latitude and temperature, while $\delta^{13}C_{org}$ signatures were only significantly correlated to DIC. These data suggest that high latitude rhodoliths rely more on $CO_2$ as an inorganic carbon source, while low latitudes rhodoliths likely take up $HCO_3^-$ directly, but none of our specimens had $\partial^{13}C_{org}$ signatures
less than -30, suggesting that none of them relied solely on diffusive $CO_2$ uptake."

Page 2, line 5:

We have rephrased the statement to: "Therefore, the ratio of stable carbon isotopes ($\partial^{13}C$) in macroalgal tissue can be used as an indicator of whether or not $HCO_3^-$ is being used (Raven et al., 2002)"

Page 2, line 26:

As mentioned above, this made our results more comparable to studies that investigated using coralline algae stable isotopes as marine proxies.

Page 3, line 8 onewards:

More specific information on the experimental materials has been added. Because we removed the section on the carbon use model, we also removed the statement on total alkalinity analysis. The text now reads "The incubations were conducted during a 24 – 70 hour light cycle with 35 µmol photons $m^{-2}$ $s^{-1}$ at 15°C, 20°C or 4°C (for the Irish, Canary Islands, and Greenland samples, respectively) in natural seawater (34 psu) in 200 - 300 ml glass or plastic jars, depending on the size of
the specimens. The duration of the light exposure depended on the specimens. The Greenland specimens were incubated longer due to their slower metabolic rates. The seawater was vacuum filtered using 0.22 µm Durapore membrane filters (Merck Millipore, Darmstadt, Germany). The jars contained stir bars and were placed on a multi-position stir plate. The rhodoliths were held on custom-made stands inside the jars to allow space for the stir bars below. The start and finish pH values (NBS) were recorded using a Sentix 51 pH electrode with an integrated temperature probe connected to a WTW 3110

pH meter (Weilheim, Germany). The seawater used during the incubation was left open to the ambient air for re-equilibration to make sure the change in pH was due to the metabolism of the algae."

Fig 4 (now Fig 5) – the legend now includes the abbreviated species names

Fig 5 (now Fig 6) – the units have been changed to µmol Kg SW-1 and we have added a map of surface ocean temperature with the mean d13C and d13Corg of Lithothamnion spp. plotted at their respective collection sites.

Fig 7 – panel labeling has been added

Comment regarding DIC recycling: Thank you for that interesting point. We have added a statement in the discussion to reflect this possibility (Page 7, lines 10-14):

"Our data suggest that calcification in northeast Atlantic rhodoliths is strongly influenced by respiratory $CO_2$, but there could additionally be recycling of DIC from precipitated carbonate material that is re-dissolved in Arctic rhodoliths. If more DIC from re-dissolved carbonate is recycled in deep Arctic rhodoliths than in shallow specimens, that could explain the higher $\partial^{13}C$ signatures we observed at 25 and 40 m compared to 11 m."

Page 5, Line 37: we have changed the wording to "directly take up $HCO_3^-$"

Page 6, Line 30 - The reference to this experiment has been removed, since we have also removed the data from the supplementary, and because we did not separate new growth from old growth.

Page 7, Line 2 - The link between DOC & $\partial13C$ has been more clearly explained by this statement: "In fact, there may be a relationship between $\partial^{13}C$ signatures and dissolved organic carbon (DOC) availability in rhodolith beds, since rhodolith food webs depend strongly on external inputs of organic matter (Grall et al. 2006, Gabara 2015), and the biogeochemical cycling within the rhodolith bed food web influences isotopic signatures."

Cornwall et al. 2015 has been added to the reference list

We have also added a reference to the work by Lee & Carpenter 2001 in the discussion:

"The positive linear relationship between $\partial^{13}C_T$ and $\partial^{13}C_{org}$ across all specimens suggests that there is a strong use of respiratory $CO_2$ during calcification in northeast Atlantic rhodoliths (Lee and Carpenter, 2001)."

**Response to reviewer 2**

The authors would like to express their gratitude to the second reviewer for their thoughtful and constructive comments. Our response to his/her comments are below.

General comments:

1. We have added a figure (Figure 2 - see attached) to show the direct relationship between d13C signatures and latitude for our samples. We also added a map showing our sites and the corresponding mean d13C signatures, based on suggestions from the first reviewer (Fig 6).

2. We have added a description of the methods for d13C and d15N analysis used by the UC Davis stable isotope facility.

"The samples were analyzed for 13C and 15N isotopes using a PDZ Europa ANCA-GSL elemental analyzer interfaced to a PDZ Europa 20-20 isotope ratio mass spectrometer (Sercon Ltd., Cheshire, UK). Samples were combusted at 1000°C in a reactor packed with chromium oxide and silvered copper oxide. Following combustion, oxides were removed in a reduction reactor (reduced copper at 650°C). N2 and CO2 were separated on a Carbosieve GC column (65°C, 65 mL/min) before entering the IRMS. During analysis, samples were interspersed with several replicates of different laboratory standards (glutamic acid and peach leaves), which were previously calibrated against NIST Standard Reference Materials.

3. Based on similar comments from the first reviewer, our light exposure experiment has been removed from the manuscript and we have improved the description of the methods and presentation of data from our pH drift experiments. We have removed the section on the carbon use model, considering that, as the first reviewer mentioned, the pH change was much lower than would be expected, and it is possible that a 24 hour was not long enough to reach the pH compensation point. We address this in a new version by stating "pH drift experiments showed that the seawater pH actually decreased after a 24 hour light incubation for both Phymatolithon calcareum and P. lusitanicum (Figure 9). However, these samples were small, and it is possible that the incubation period was not long enough to detect a significant change in pH."

Minor Comments:

1. CCMs are now defined at the beginning of the abstract

2. We have edited the symbols so that $\partial$ is used throughout

3. The numbers behind the parenthesis represent lat/long coordinates

Added References:

Pardo, C., Lopez, L., Peña, V., Hernández-Kantún, J., Le Gall, L., Bárbara, I. and Barreiro, R.: A Multilocus Species Delimitation Reveals a Striking Number of Species of Coralline Algae Forming Maerl in the OSPAR Maritime Area, edited by D. Fontaneto, PLoS One, 9(8), e104073, doi:10.1371/journal.pone.0104073, 2014.

**List of Relevant Changes**

• Major changes and additions made to the Materials and Methods based on reviewer comments (particularly regarding pH shift experiments)
• Addition of Figure 6 (map) and Figure 9 (pH compensation points) based on comments from reviewer 1 and resulting additional experiments conducted to obtain pH compensation points for more species
• Addition of Figure 2 (relationship between $\partial$13C and latitude) based on comments from reviewer 2

• Added statements to the discussion based on reviewer comments/suggestions
Below you can find the marked-up version of the manuscript

**Latitudinal trends in stable isotope signatures and carbon concentrating mechanisms of northeast Atlantic rhodoliths**

Laurie C. Hofmann[1], Svenja Heesch[2]

[1]Microsensor Group, Max Planck Institute for Marine Microbiology, Bremen, 28359, Germany
[2]Centre National de Recherche Scientifique, UMR 8227, Station Biologique de Roscoff, Roscoff, 29680, France

*Correspondence to*: Laurie C. Hofmann (lhofmann@mpi-bremen.de)

**Abstract.** Rhodoliths are free-living calcifying red algae that form extensive beds in shallow marine benthic environments (< 250 m), which provide important habitats and nurseries for marine organisms and contribute to carbonate sediment accumulation. There is growing concern that these organisms are sensitive to global climate change, yet little is known about
their physiology. Considering their broad distribution along most continental coastlines, their potential sensitivity to global change could have important consequences for the productivity and diversity of benthic coastal environments. The goal of this study was to determine the plasticity of  carbon concentrating mechanisms (CCMs) of rhodoliths along a latitudinal gradient in the Northeast (NE) Atlantic using natural stable isotope signatures. The $\delta^{13}C$ signature of macroalgae can be used to provide an indication of the preferred inorganic carbon source ($CO_2$ vs. $HCO_3^-$). Here
we present the total ($\delta^{13}C_T$) and organic ($\delta^{13}C_{org}$) $\delta^{13}C$ signatures of NE Atlantic rhodoliths with respect to changing environmental conditions along a latitudinal gradient from the Canary Islands to Spitsbergen. The $\delta^{13}C_T$ signatures (-11.9 to -0.89) of rhodoliths analysed in this study were generally higher than the $\delta^{13}C_{org}$ signatures, which ranged from -25.7 to -2.8. We observed a decreasing trend in $\delta^{13}C_T$ signatures with increasing latitude and temperature, while $\delta^{13}C_{org}$ signatures were only significantly correlated to DIC. These data suggest that high latitude rhodoliths rely more on $CO_2$ as an inorganic
carbon source, while low latitudes rhodoliths likely take up $HCO_3^-$ directly, but none of our specimens had $\partial^{13}C_{org}$ signatures less than -30, suggesting that none of them relied solely on diffusive $CO_2$ uptake. However, depth also has a significant effect on both skeletal and organic $\delta^{13}C$ signatures, suggesting that both local and latitudinal trends influence the plasticity of rhodolith inorganic carbon acquisition and assimilation. Our results show that many species, particularly those at lower latitudes, have carbon concentrating mechanisms that facilitate $HCO_3^-$ use for photosynthesis. This is an important
adaptation for marine macroalgae, because $HCO_3^-$ is available at higher concentrations than $CO_2$ in seawater, and this becomes even more extreme with increasing temperature. The flexibility of CCMs in northeast Atlantic rhodoliths observed in our study may provide a key physiological mechanism for potential adaptation of rhodoliths to future global climate change.

Copyright statement: The authors have read and agree to the copyright terms outlined by the journal.

[revised manuscript text omitted]